# Computational and experimental exploration of statin and statin-like compounds as potential treatment of schistosomiasis

Kehinde F. Paul-Odeniran[1,2]*, Paul O. Odeniran[3], Cécile Häberli[4,5], Jennifer Keiser[4], Charles A. Laughton[1]*

1 Centre for Biomolecular Sciences, School of Pharmacy, University of Nottingham, University Park, Nottingham, United Kingdom, 2 Department of Natural Sciences, Faculty of Pure and Applied Sciences, Precious Cornerstone University, Oyo, Nigeria, 3 Department of Veterinary Parasitology and Entomology, Faculty of Veterinary Medicine, University of Ibadan, Oyo, Nigeria, 4 Department of Medical Parasitology and Infection Biology, Swiss Tropical and Public Health Institute, Allschwil, Switzerland, 5 University of Basel, Basel, Switzerland

* folu.paul-odeniran@nottingham.ac.uk (KFP-O); charles.laughton@nottingham.ac.uk (CAL)

## Abstract

Schistosomiasis remains a global health challenge, affecting over 240 million people each year. Current treatment options, including praziquantel, are limited by their inability to target immature schistosomula and growing concerns around drug resistance. Targeting 3- hydroxy 3-methylglutaryl coenzyme A reductase from *Schistosoma mansoni* (*Sm*HMGR), a key enzyme in the parasite's mevalonate pathway, presents a promising therapeutic strategy. Although the antischistosomal potential of statins has been previously explored, this study introduces a novel integrated pipeline that combines homology modelling, large scale analogue screening of a $10^{13}$ compound chemical space, validated molecular docking, molecular dynamics simulations, and experimental validation to systematically identify *Sm*HMGR inhibitors. Following computational prioritisation, experimental validation revealed modest activity of three commercially available statins (lovastatin, pravastatin, and pitavastatin) against schistosomula with up to 56.3% at 50 µM, but no significant time-dependent effects. Three novel analogues of pitavastatin exhibited enhanced schistosomicidal activity, revealing activities of up to 96% and 55% against schistosomula and adult worms at 50 µM, respectively. These findings highlight the potential of structural modifications to improve the efficacy of statin-based compounds against *S. mansoni*.

## Author summary

Schistosomiasis is a neglected tropical disease that affects more than 240 million people, mainly in low-resource settings. The disease is currently treated with a single drug, praziquantel, which is not effective against early-stage parasites

**Data availability statement:** All data generated and validations are referenced in both the manuscript and supplementary files and results have been processed for the post-molecular dynamics analysis.

**Funding:** This study was funded by the United Kingdom Royal Society (Grant number: NIF/R1/211767) awarded to KFP-O. The funders had no role in study design, data collection and analysis, decision to publish, or preparation of the manuscript. KFP-O received a salary during this study period through the Royal Society grant. No other authors received salaries from the funder.

**Competing interests:** The authors have declared that no competing interests exist.

and may face resistance in the future. In this study, we used computational and experimental methods to identify structural analogues of statins for potential re-purposing as new treatments for schistosomiasis. Targeting the parasite enzyme HMGR, which plays a vital role in its survival, we applied molecular modelling, docking, and molecular dynamics simulations to prioritise promising compounds. These statin analogues were then tested in laboratory assays, where several showed improved activity against both larval and adult stages of the parasite compared to the original statin compounds. Our findings support the potential of rationally designed statin analogues as affordable and accessible therapeutic candidates for schistosomiasis, particularly in endemic regions with limited treatment options.

## 1. Introduction

Schistosomiasis affects over 240 million individuals worldwide, with more than 700 million at risk of infection, highlighting its substantial global burden [1]. The disease is endemic in 78 low- and middle-income countries (LMICs) across tropical and sub-tropical regions, where it disproportionately affects vulnerable populations in areas with limited access to clean water, sanitation, and healthcare [2–4]. *Schistosoma mansoni*, S. *japonicum*, and S. *haematobium* are the three dominant species responsible for human schistosomiasis, transmitted through contact with freshwater contaminated by infected snails [5,6].

While praziquantel (PZQ) remains the mainstay of schistosomiasis treatment, its efficacy against juvenile parasites is limited and variable. Although in vitro studies have shown that PZQ can kill schistosomula under specific conditions, it is generally less effective against early developmental stages *in vivo*, contributing to concerns about reinfection and treatment gaps in endemic populations [7–10].

This limitation leads to reinfection risks and necessitates repeated treatments. Moreover, the extensive use of praziquantel raises concerns about potential drug resistance, underscoring the need for alternative therapeutic approaches [11–13]. The lack of a licensed vaccine further hampers control efforts, which rely on mass drug administration (MDA) campaigns in endemic regions. While MDA programs have reduced disease prevalence, reinfection remains a persistent issue [14–17]. To reduce the global burden, there is an urgent need for innovative therapies targeting multiple parasite stages, alongside strengthened public health infrastructure and improved water and sanitation systems [4,18].

The mevalonate pathway is essential for the survival and development of *Schistosoma mansoni* [19–21]. It plays a key role in sterol biosynthesis, regulating the production of lipids critical for the parasite's growth and reproduction [22]. A central enzyme in this pathway, 3-hydroxy-3-methylglutaryl-coenzyme A reductase (HMGR), catalyzes the rate-limiting step and serves as a critical metabolic regulator [23]. S. *mansoni* HMGR (*Sm*HMGR), is crucial for producing non-sterol lipids that stimulate egg production, linking it directly to the parasite's reproductive success. Validated as

a therapeutic target, SmHMGR offers significant potential for selective drug development. Its indispensable role in parasite survival and reproduction makes it a promising focus for novel therapies [19,22,24–26].

Statins, widely recognized for their cholesterol-lowering effects through inhibition of human HMGR, have attracted attention for their potential to target SmHMGR [19,27]. Previous studies have suggested the efficacy of some statins in inhibiting SmHMGR and proposed their potential as treatments for schistosomiasis, highlighting the need for further exploration in this area [28].

This study was designed as an exploratory investigation especially into the potential of structural analogues of statins to inhibit SmHMGR. The focus was on systematically evaluating the binding profiles of these compounds using a structure-based computational pipeline. Homology modelling, molecular docking, and binding free energy estimations were employed to construct a high-quality structural model of SmHMGR and prioritise analogues based on predicted interaction strength and specificity.

In parallel, selected compounds were evaluated in biological assays to preliminarily assess their schistosomicidal activity and support computational predictions. By establishing this integrated computational-experimental workflow, the study aims to generate foundational insights into the chemical space surrounding statins in the context of SmHMGR inhibition. This framework is intended to guide future medicinal chemistry efforts and compound selection, contributing to the rational design of selective inhibitors for schistosomiasis therapy.

## 2. Methodology

### 2.1. Ethics

Laboratory studies were carried out in accordance with Swiss national and cantonal regulations on animal welfare at the Swiss Tropical and Public Health Institute (Basel, Switzerland; permission no. 545).

### 2.2. Homology modelling of *Schistosoma mansoni* HMGR: Sequence analysis and model construction

The homology modelling of SmHMGR was initiated with sequence analysis of both the template and the target protein. SmHMGR sequence (UniProt ID: P16237) [20] was obtained from the UniProt database [29]. The sequence of the catalytic domain of human HMGR (UniProt ID: P04035) and its corresponding 3D structure (PDB ID: 1HWK), co-crystallized with atorvastatin, a moderate inhibitor of SmHMGR was retrieved [28,29]. A search of the Protein Data Bank (PDB) was conducted to identify available crystal structures of HMGR from non-human organisms; however, no suitable high-resolution structures were found that offered better alignment or structural completeness than the human HMGR. Therefore, the human protein was selected as the most appropriate template. Following this, a sequence alignment was performed using Clustal Omega to compare the catalytic portion of SmHMGR sequence with the human HMGR sequence, focusing on conserved regions vital for enzyme function, particularly in the catalytic domains [30,31].

Homology modelling of SmHMGR was then conducted using MODELLER [32], with the human HMGR structure (PDB ID: 1HWK) serving as the template, given its high sequence similarity to SmHMGR. Though 1HWK is a tetrameric structure, the dimeric form (chains A and B) was used as the template for modelling, as the human HMGR is catalytically active in this dimeric form [33]. This dimer features bipartite active sites, where residues from neighbouring monomers contribute to the active site.

Attention was given to ensuring the correct formation of these bipartite active sites, along with the overall structural accuracy of the dimer. Specifically, chain A was used to generate 140 monomers, and chain B was also used to generate 140 monomers. After monomer generation, the dimers were assembled by pairing each monomer from chain A with its corresponding monomer from chain B. For example, model 1 of chain A was assembled with model 1 of chain B. This process ensured an accurate representation of the monomer-monomer interface and the bipartite active sites in the final dimer models.

## 2.3. Inverse docking of atorvastatin to *Sm*HMGR models

To identify the most suitable model of *Sm*HMGR from a set of 140 homology models, we performed inverse docking with atorvastatin as the ligand. This approach is grounded in validated methodologies for inverse docking, which have been widely employed to identify favourable receptor conformations or potential targets for known ligands [34,35]. In our study, it enabled the selection of receptor conformers that demonstrated high compatibility with atorvastatin, thereby streamlining subsequent virtual screening steps.

The preparation involved ensuring that atorvastatin and each *Sm*HMGR model were converted to pdb files. AutoDock Vina was utilised for the docking simulations, with the workflow managed using a Python-based system. Python packages including mdtraj, numpy, and dask_jobqueue facilitated data handling and parallel processing. The grid was centred on the active site of *Sm*HMGR at coordinates X = 19.619 Å, Y = 8.0145 Å, and Z = 15.60 Å, with dimensions of 17.92 Å (X-axis), 17.49 Å (Y-axis), and 15.41 Å (Z-axis). An exhaustiveness value of 30 was applied to ensure a thorough search of the conformational space. More than binding modes per ligand were generated. The energy range was set to 3 kcal/mol, retaining all poses within this energy window from the top-ranked conformation. Default scoring functions and search parameters were used unless otherwise specified. Post-docking, mdtraj was used to process and analyse the results, focusing on both docking scores and binding poses. Binding quality was initially evaluated based on docking scores, with lower scores indicating stronger predicted interactions. In addition, the top-ranked ligand poses were visually inspected using Maestro [36] and Discovery Studio [37] to confirm correct orientation within the *Sm*HMGR binding pocket and to assess key interactions such as hydrogen bonding and hydrophobic contacts. Models demonstrating both favourable docking scores and plausible binding orientations were selected for further validation.

## 2.4. Structural validation method

To assess the structural integrity of the top-performing homology models, several key metrics were calculated, including clashscore, MolProbity score, Ramachandran plot analysis, rotamer outliers, and QMEAN score [38–41]. These evaluations were performed using the structural assessment plugin available through the Swiss-Model server [42]. The clashscore and MolProbity score provided insights into stereochemical quality, while Ramachandran plots were used to analyse the backbone dihedral angles. Rotamer analysis was conducted to check for side-chain conformations, and QMEAN scores were calculated to assess overall model reliability.

## 2.5. Analogue Hunting, prefiltering and principal component analysis

Infinisee 5.0, an integrated ligand database with six chemical spaces containing approximately $10^{13}$ diverse molecules, served as the primary resource for identifying statin-like compounds [43]. The chemical spaces within Infinisee 5.0 include REAL Space, CHEMriya, eXplore, Freedom Space, Knowledge Space, and GalaXi. All spaces contain synthesizable and accessible compounds, except Knowledge Space, which is predominantly theoretical. To identify statin analogues, parent compounds, including atorvastatin, cerivastatin, fluvastatin, lovastatin, pravastatin, and rosuvastatin, were uploaded individually to the Infinisee database. The ECFP4 fingerprint search was employed, with a minimum similarity threshold of 0.7, leading to the identification of thousands of analogues. Given the varied properties of these analogues, a secondary screening process was applied using the Analyzer tab within Infinisee. This filtering focused on drug-like, lead-like, and fragment-like characteristics. The parameters for filtering included:

- Drug-likeness: No more than two violations of the following criteria: MW ≤ 500 Da, LogP ≤ 5, Rotatable bonds ≤ 10, HBA ≤ 10, HBD ≤ 5 [44].

- Fragment-likeness: MW ≤ 300 Da, LogP ≤ 3, HBA ≤ 3, HBD ≤ 3 [45].

- Lead-likeness: MW < 450 Da, LogP < 4 [46].

Physicochemical properties of the parent statins were determined using Swiss-ADME [47], which helped assess conformity to Lipinski's Rule of Five (Ro5) for drug-likeness, as well as the Rule of Three (Ro3) for fragment-likeness, and lead-likeness criteria. Principal component analysis was performed on the physicochemical descriptors of the statin analogues using the scikit-learn library in Python. Before PCA, the data were standardised (zero mean and unit variance) using the StandardScaler function to ensure comparability across variables. The first two principal components were used to visualise the chemical diversity and separation of filtered and unfiltered compound sets.

### 2.6. Validation of docking protocol through ligand enrichment

To establish a validated protocol for docking statin analogues to *Sm*HMGR, we first needed to identify the most effective docking algorithm for distinguishing between active inhibitors of HMGR and decoy compounds. A ligand enrichment protocol was employed using AutoDock Vina, Glide SP, and Glide XP algorithms [48,49]. The performance of each algorithm was evaluated using enrichment factor (EF) and receiver operating characteristic (ROC) curves [50]. The dataset for this enrichment study was sourced from the DUD-Z database, comprising 43 known active inhibitors of HMGR and 1893 decoy compounds [51]. We selected these three specific algorithms to compare the performance of both open-source and commercial docking software. AutoDock Vina was chosen as it is an open-source tool widely used in molecular docking studies due to its efficiency and accessibility. Glide SP and Glide XP, on the other hand, are part of the commercial Schrödinger Suite and are known for their high precision and advanced scoring functions, providing a good basis for comparison with open-source software.

**2.6.1. Active and decoy ligand preparation.** A total of 43 active HMGR inhibitors and 1893 decoy compounds were downloaded in structure-data file (SDF) format. Ligand preparation was performed using LigPrep in Schrödinger Suite 2023 [52]. Each ligand was processed to ensure the assignment of correct bond orders, and stereoisomers were generated where necessary. The OPLS3e force field was applied to the three-dimensional geometries of all ligands. Ionisation states were assigned using Epik 3.6 at a physiological pH of 7.4 [53]. To account for stereochemical variability in cases where chiral centres were undefined, an average of 10 stereoisomers per ligand was generated.

**2.6.2. Protein preparation and docking protocols.** For the protein target, we used the *Sm*HMGR modelled and validated structure from previous steps. The protein was prepared using the Protein Preparation Wizard in Schrödinger [54], adjusting bond orders and adding missing hydrogen atoms. Protonation states for ionisable residues were adjusted, and the structure underwent restrained energy minimisation with heavy atom restraints applied, using default parameters: a convergence threshold of 0.3 kcal/mol/Å and a maximum of 2500 steps.

For the docking of the actives and decoys with AutoDock Vina, the grid box dimensions for this protocol were the same as those used in the inverse docking experiments with atorvastatin, which were employed to select the best *Sm*HMGR model in section 2.2.

For Glide, the receptor grid was centred on the cognate ligand, atorvastatin, with the grid box dimensions extended by 15 Å in all directions (X, Y, and Z) to fully cover the active site. The van der Waals radii of the ligand atoms were scaled by 0.8, and a partial charge cutoff of 0.25 was applied to ensure a thorough exploration of binding interactions. High-throughput virtual screening (HTVS) was initially performed using the standard precision (SP) scoring function to enable rapid and efficient screening of the ligand library. Ligands were treated with full flexibility during the docking, with no post-docking minimisation applied. In addition to the SP mode, docking was also conducted independently using the extra precision (XP) mode for a more detailed analysis.

**2.6.3. ROC curve generation.** After the docking simulations were complete, the docking scores for both the active inhibitors and decoys were obtained and ranked. The performance of each docking algorithm was then validated using ligand enrichment metrics based on these ranked scores. To evaluate the performance of each algorithm, the enrichment factor (EF) was calculated at multiple ranked percentages: 1%, 5%, 10%, 15%, 20%, 25%, 50%, and 100% of the dataset. The EF indicates how well the docking algorithm enriches for active inhibitors at the top of the ranked list compared to

a random distribution. Additionally, ROC curves were plotted for each algorithm, representing the trade-off between the true positive rate (sensitivity) and the false positive rate (1-specificity). The area under the curve (AUC) was calculated to assess each algorithm's overall performance. Through these combined metrics, we identified which algorithm most effectively distinguished between active HMGR inhibitors and decoy compounds. The validated protocol derived from this analysis was then used to guide the docking of statin analogues in subsequent experiments.

## 2.7. Molecular docking of statin analogues and their parent compounds

The selected prefiltered statin analogues and parent statins were prepared according to the ligand preparation methodology outlined above. Similarly, the receptor grid was generated following the previously described steps. These were docked with the algorithm that emerged as the most suitable algorithm for statin docking based on the enrichment factor and AUC performance into the active site of the *Sm*HMGR model. The docking results are expressed as Glide scores, where more negative values indicate stronger predicted binding affinities. These scores were used to assess both the parent statins and their analogues, identifying the most promising candidates to proceed into MD simulation.

## 2.8. Molecular dynamics of statin analogues using AMBER

The system was prepared for molecular dynamics (MD) simulations using the AMBER22 suite [55]. The parent statin and statin analogues identified from the docking process were parameterised using the GAFF2 force field to ensure accurate treatment of the ligands [56]. The ligand-receptor complex was solvated in a rectangular box of TIP3P water molecules, with a 15 Å buffer around the complex to ensure complete solvation [57]. The AMBER ff19SB force field was employed for the receptor [58], while counterions were added to neutralise the system. Energy minimisation was carried out in several stages to relieve any steric clashes. Initially, steepest descent and conjugate gradient minimisation were applied to the solvent molecules while restraining the heavy atoms of the solute with a force constant of 500 kcal/mol·Å$^2$ [59]. In the next phase, the restraints were reduced to 100 kcal/mol·Å$^2$ to allow more flexibility, followed by a minimisation step where the restraints were decreased to 5 kcal/mol·Å$^2$. A final minimisation was performed with no restraints to fully relax the system [55]

The system was heated gradually from 0 K to 300 K under constant volume (NVT) conditions with a restraint of 50 kcal/mol·Å$^2$ applied to the solute [59]. Following the heating phase, equilibration was performed under constant pressure (NPT) conditions at 1 atm, with the restraint force constants progressively reduced from 50 kcal/mol·Å$^2$ to 10 kcal/mol·Å$^2$, and then to 2 kcal/mol·Å$^2$. Finally, a no-restraint equilibration was done to allow full relaxation of the system at 300 K and 1 atm. The production phase of the MD simulation was conducted under NPT conditions without restraints. The system was simulated for 500 nanoseconds (ns). To assess reproducibility and capture diverse conformational outcomes, we performed duplicate simulations using unique randomised starting velocities. Long-range electrostatics were treated using the particle mesh Ewald (PME) method, and all bonds involving hydrogen atoms were constrained using the SHAKE algorithm [55].

Following MD simulations, post-MD analysis and binding free energy calculations were performed to evaluate ligand–receptor stability and affinity.Trajectories, including root mean square deviation (RMSD), were analyzed using CPPTRAJ and PTRAJ module [60]. Visualization and data plotting were performed with Origin analytical tool [61]. Thermodynamic analysis was carried out using the Molecular Mechanics Generalised Born Surface Area (MMGBSA) method to estimate the binding free energy of each statin analogue within the SmHMGR binding site [62]. The binding free energies of the statin analogues were computed using the MMGBSA.py script integrated within Amber22. This script analyzes binding free energies from MD simulation snapshots by employing continuum solvent models. For the calculations, snapshots were extracted from the MD trajectories to provide a detailed assessment of the binding affinity.

The binding free energy ($\Delta G_{bind}$) is calculated as follows:

$$\Delta G_{bind} = G_{complex} - (G_{receptor} + G_{inhibitor}) \qquad (1)$$

$$\Delta G_{bind} \ = \ \Delta G_{gas} \ + \ \Delta G_{sol} \ - \ T\Delta S \tag{2}$$

$$\Delta G_{gas} \ = \ \Delta E_{int} \ + \ \Delta E_{ele} \ + \ \Delta E_{vdW} \tag{3}$$

$$\Delta G_{sol} \ = \ \Delta G_{PB} \ + \ \Delta G_{np,sol} \tag{4}$$

The interactions contributing to gas-phase energy ($\Delta G_{gas}$) include internal forces ($\Delta E_{int}$), electrostatic interactions ($\Delta E_{ele}$), and van der Waals forces ($\Delta E_{vdW}$) [63]. Meanwhile, solvation-free energy ($\Delta G_{sol}$) arises from both polar ($\Delta G_{PB}$) and non-polar ($\Delta G_{np,sol}$) solvation contributions.

### 2.9. Experimental Validation

**2.9.1. Compounds and culture media.** A total of 70 compounds underwent molecular dynamics simulations and MMGBSA analysis to calculate their binding affinities. From these, nine statin-like analogues, including the parent statins with high binding scores, were selected for *in vitro* studies. These compounds were synthesized by LGC-Standard (Germany) and subsequently delivered to the Swiss Tropical and Public Health Institute (Basel, Switzerland) for testing against *Schistosoma mansoni* newly transformed schistosomula (NTS) and adult *S. mansoni*. Nine of the compounds were dissolved in pure DMSO at a concentration of 10 mM, while three were dissolved in methanol at a concentrations of 10 mM. The resulting stock solutions were further diluted using culture medium.

For *in vitro* assays and incubation of NTS, Medium 199 (Gibco, Waltham, MA, USA) was supplemented with 1% penicillin-streptomycin solution (10,000 U/mL penicillin, 10 mg/mL streptomycin; Bioconcept AG, Allschwil, Switzerland) and 5% (v/v) inactivated horse serum (Gibco, Waltham, MA, USA). For adult *S. mansoni*, RPMI 1640 medium (Gibco, Waltham, MA, USA) was similarly prepared by adding 1% penicillin-streptomycin solution and 5% (v/v) horse serum.

**2.9.2. Parasites.** Cercariae of S. *mansoni* (Liberian strain) were obtained from *Biomphalaria glabrata* snails infected with S. *mansoni* and prepared for transformation following established protocols (adapted from [64]. Snails were placed individually in 24-well plates and exposed to bright light (e.g., neon lamp at 36 W, 4000 K) for 3–4 hours to induce cercarial shedding. The released cercariae were collected, filtered to remove debris, and transformed into NTS by mechanical tail separation. This transformation was achieved by repeated passage through syringes fitted with Luer-Lok tips. After tail detachment, NTS were washed in Hanks' balanced salt solution (HBSS; Gibco, Waltham, MA, USA) supplemented with 1% penicillin-streptomycin and then suspended in M199 culture medium. They were incubated overnight at 37°C in a humidified 5% $CO_2$ incubator. Adult S. *mansoni* were harvested from the hepatic portal system and mesenteric veins of mice infected with approximately 100 cercariae 50 days earlier. The worms were washed in phosphate-buffered saline (PBS, pH 7.4; Sigma-Aldrich, Buchs, Switzerland) supplemented with 1% penicillin-streptomycin and maintained in RPMI 1640 culture medium at 37°C in a 5% $CO_2$ incubator until use.

**2.9.3. *In vitro* phenotypic drug sensitivity assays.** Phenotypic drug sensitivity assays were conducted to evaluate the activity of test compounds on both NTS and adult S. *mansoni*. For NTS, assays were performed in 96-well plates (Sarstedt, Nümbrecht, Germany), with approximately 50 NTS per well. Adult worms (at least three of both sexes) were placed in 24-well plates (Sarstedt, Nümbrecht, Germany). Test compounds were added to the respective culture media at a concentration of 10 µM; if no significant activity (≥66%) was observed, assays were repeated at 50 µM. Parasites were incubated at 37°C with 5% $CO_2$ for 72 hours. Daily phenotypic assessments of adult worms were performed under a microscope (Carl Zeiss, Germany; magnification 10–40×). Parameters such as motility, viability, and morphological changes were scored using a pre-defined scale. NTS viability was assessed after 72 hours using similar criteria [64].

## 3. Result and discussion

### 3.1. Sequence alignment and homology models construction

Sequence analysis revealed a 52.33% similarity between the catalytic domains of human HMGR and *Sm*HMGR, with conserved binding site residues highlighted in yellow (Fig 1). This degree of conservation underscores the functional importance of *Sm*HMGR and its suitability for homology modeling. Among the 140 homology models generated for *Sm*HMGR, careful evaluation of structural integrity around both the global fold and the active site region was carried out to ensure accuracy and reliability. Particular attention was paid to the preservation of conserved residues within the catalytic and dimerization domains, as any misalignment or distortion in these regions could compromise the reliability of downstream docking. Approximately 40 models were excluded due to issues such as disrupted secondary structure elements, poor

**Fig 1. Sequence alignment of the catalytic domains of human HMGR and *Sm*HMGR.** Conserved residues are highlighted in yellow. Asterisks (*) indicate identical residues, colons (:) represent conserved substitutions, and periods (.) indicate semi-conserved substitutions.

loop modelling near the active site, or steric clashes that would hinder ligand binding. The final selected model, illustrated in Fig 2 and selected through a process detailed in a later section, exhibited close structural alignment with the human HMGR template, preserving key alpha-helices, beta-sheets, and the spatial configuration of catalytic residues. This preservation is essential for maintaining the enzyme's catalytic core, which relies on residues contributed by both monomers. Accurate dimer assembly ensures proper alignment of these residues, supporting functional bipartite active sites and maintaining the quaternary structure.

The dimeric interface, highlighted in red in Fig 2, further validates the structural fidelity of the model, as residues from both monomers effectively support the catalytic core. This high degree of conservation suggests the functional viability of the modeled *Sm*HMGR, establishing its suitability for downstream computational and experimental analyses.

### 3.2. Inverse docking experiments and model selection

The primary objective of inverse docking is to identify receptor models that best accommodate a known ligand, providing insights into the receptor's suitability for further structural studies. Inverse docking was applied to evaluate the retained homology models of *Sm*HMGR, with atorvastatin, a moderate inhibitor of *Sm*HMGR, used as the probe ligand. This process enables the identification of receptor models that closely mimic the native binding environment of the ligand as it compares with the template, thus helping to select the models with the highest potential for accurate structural validation.

Docking simulations were performed using AutoDock Vina and yielded scores ranging from -3.4 to -8.3 kcal/mol, with lower scores indicating stronger binding affinities. Most models clustered within the -6.5 to -7.5 kcal/mol range (Fig 3). Several models demonstrated favourable interactions with atorvastatin, and the top three; recA, recB, and recC were shortlisted based on their high predicted binding affinities. Among these, recA achieved the best docking score of −8.3 kcal/mol, closely followed by recB and recC, with scores of −8.1 and −8.0 kcal/mol, respectively. The superior binding

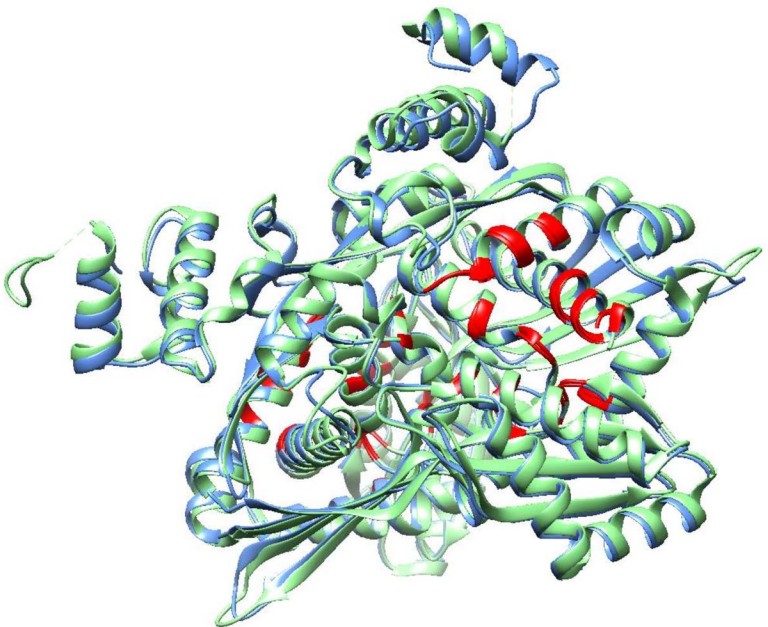

**Fig 2. Structural alignment of the human HMGR crystal structure (light green) and the *Sm*HMGR homology model (cornflower blue), highlighting the dimer interface residues in red.** These residues are contributed by both monomers and are spatially positioned to support the catalytic core. The conservation of this interface underscores the importance of accurate dimer assembly in maintaining enzymatic function.

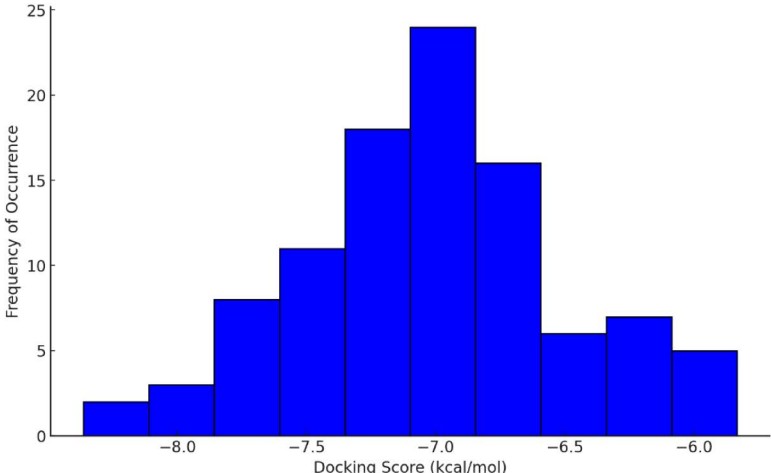

**Fig 3. Distribution of docking scores for receptor models.**

energy and favourable interaction profile of recA reinforced its selection as the optimal model for subsequent computational and experimental studies.

The binding pose of atorvastatin within recA (Fig 4A) was visualized and compared to the template structure 1HWK (Fig 4B and 4C). In recA, atorvastatin establishes key hydrogen bonds and hydrophobic interactions with residues in the active site, closely mirroring the interactions observed in the template structure. Specifically, residues such as Lys691, Lys692, Arg590, Ser661, Ser565 and Asp690 play significant roles in stabilizing atorvastatin in the binding pocket of 1HWK [33] and this is replicated by Lys233, Lys234, Arg132, Ser203, Ser107 and Asp232 in recA further supporting it as a reliable model. The high degree of similarity between the binding poses of atorvastatin in recA and the template structure reinforces recA as a robust homology model for *Sm*HMGR docking studies.

### 3.3. Structural quality assessment and model selection

Despite recA emerging as the top performing model based on its docking score and ligand interactions, the other two shortlisted models (recB and recC) also demonstrated strong potential and are therefore included in the validation step. Including these additional models ensures that subtle but meaningful structural variations are not prematurely excluded, allowing for a more comprehensive comparison of *Sm*HMGR conformations.

The structural assessment metrics across the three models (Table 1) revealed that all are of high quality and suitable for further investigation. While the numerical differences are modest, they provide insights into the individual strengths of each model. recA recorded the lowest clashscore (131.54), the highest QMEAN score (0.714), and the highest percentage of residues in favoured Ramachandran regions (94.61%), with the fewest outliers (1.51%). These results indicate strong backbone geometry and global structural integrity.

However, recA displayed slightly higher MolProbity (3.68) and rotamer outlier (7.2%) scores compared to recB (3.41 and 5.9%, respectively). Despite these minor differences, the overall performance of recA remains the most consistent and well balanced across all key structural metrics. recB and recC remain valuable for comparative validation, but recA will serve as the reference model for subsequent computational and experimental investigations.

### 3.4. Statin analogues curation and prefiltering

Although the parent statins are already approved and marketed, it was important to characterise their physicochemical properties to establish a reference framework for analogue selection. This evaluation, based on drug-likeness,

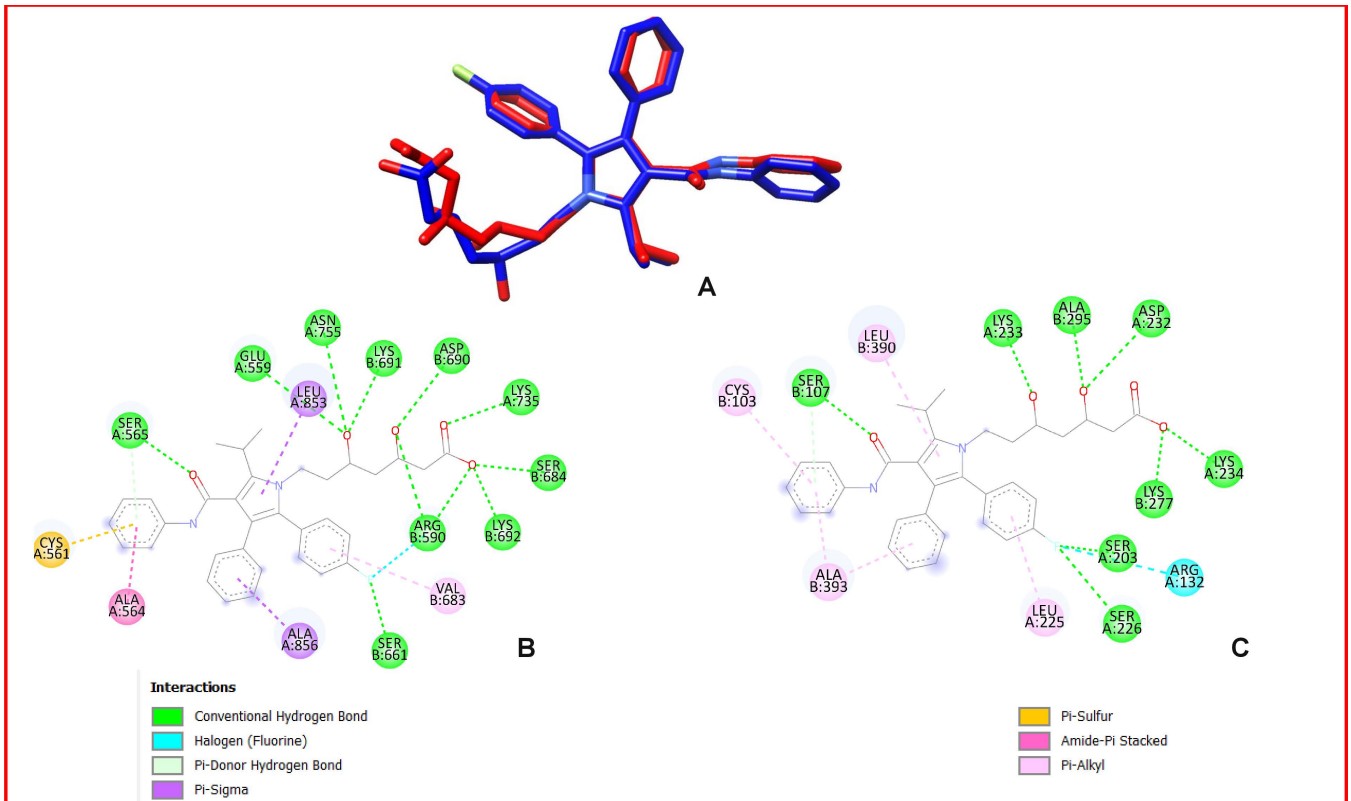

**Fig 4. A.** Binding pose of Atorvastatin in recA and the template. B and C is the protein ligand interaction in the template (1HWK) and the homology model (recA) respectively.

**Table 1. Structural validation metrics for the shortlisted *Sm*HMGR homology models (recA, recB, and recC).**

| Metric | RecA | RecB | RecC |
|---|---|---|---|
| Clash Score | 131.54 | 139.72 | 142.81 |
| MolProbity Score | 3.68 | 3.41 | 3.59 |
| Ramachandran favoured (%) | 94.61 | 93.85 | 94.02 |
| Ramachandran outliers (%) | 1.51 | 1.84 | 1.72 |
| Rotamer outliers (%) | 7.2 | 5.9 | 6.4 |
| QMEAN | 0.714 | 0.698 | 0.690 |

lead-likeness, and fragment-likeness criteria as described in section 2.3, provided a benchmark for identifying structural and physicochemical features desirable in potential analogues. Understanding how the parent statins aligned with these parameters ensured that the downstream analogue curation process was guided by rational property-based constraints. The analysis presented in Table 2 showed that all the parent statins conformed to the drug-likeness rule, except for atorvastatin, which violated more than two parameters, including molecular weight, number of rotatable bonds, and number of hydrogen bond donors.

After determining the physicochemical properties of the parent statins, each was uploaded to the Infinisee database to mine analogues using the ECFP4 fingerprint search with a similarity threshold of 0.7. Although the primary goal of this study was not to optimise fragments or lead-like compounds, fragment-likeness and lead-likeness filters were applied at

**Table 2. Physicochemical properties of parent statins.**

| Parent Statins | Molecular Weight (Da) | LogP | No. of Rotatable Bonds | No. of HBA | No. of HBD |
|---|---|---|---|---|---|
| Atorvastatin | 558.6 | 5 | 12 | 6 | 4 |
| ,Fluvastatin | 411.5 | 3.5 | 8 | 5 | 3 |
| Pravastatin | 424.5 | 1.6 | 11 | 7 | 4 |
| Pitavastatin | 459.5 | 3.6 | 11 | 7 | 3 |
| Lovastatin | 404.5 | 4.3 | 7 | 5 | 1 |
| Rosuvastatin | 481.5 | 1.6 | 10 | 10 | 3 |

this stage to characterise the chemical space of the screened library. This provided an exploratory overview of the diversity and quality of the retrieved analogues, helping to identify whether the library contained candidates falling within drug-like, lead-like, or fragment-like categories before final selection.

It is worth noting that the open-ring structure of lovastatin and pravastatin makes them analogues of each other [65]. Therefore, it is unsurprising that the analogues of lovastatin identified by Infinisee were categorized with pravastatin as the base structure. However, it was assumed that not all these analogues would be ideal candidates for drug development, as many may not meet the criteria for drug-likeness, lead-likeness, or fragment-likeness. A secondary screening step was therefore necessary to filter out compounds that did not meet these criteria as described in section 2.4 and focus on those with the most potential for further investigation.

To aid in interpreting this large dataset of analogues and their physicochemical variability, a Principal Component Analysis (PCA) was performed (Fig 5). This statistical technique allowed us to visualise the chemical diversity of the analogues in terms of their molecular weight, logP, number of rotatable bonds, hydrogen bond acceptors, and the number of rings. By reducing the complexity of the dataset, the PCA helped to identify patterns and clustering within the chemical space, which in turn highlighted the analogues that best matched the desired drug-like properties. Visualising this diversity was crucial for ensuring that the analogues selected for further study spanned a wide range of chemical space, offering greater potential for identifying promising candidates. Table 3 summarises the total number of analogues identified for each parent statin and the number that passed through the drug-like, lead-like, and fragment-like filters:

As shown in Table 3, pravastatin and fluvastatin produced the highest number of drug-like analogues, with 143 and 94 analogues, respectively. These two statins displayed favourable properties that enabled a higher proportion of their analogues to pass through the filtering process. In contrast, atorvastatin and rosuvastatin did not yield any drug-like, lead-like, or fragment-like analogues. This result is consistent with the higher molecular weights and greater molecular complexity of these compounds, which made them less likely to meet the filtering criteria. The filtering process reduced the initial pool of 11,959 analogues to 294 drug-like compounds, representing approximately 2.7% of the total output.

By focusing on the most promising candidates, this prefiltering step ensured that only analogues with favourable drug-like, lead-like, or fragment-like characteristics would proceed to the next phase of the docking studies. This refinement was crucial to enhance the efficiency and effectiveness of the docking process, reducing the likelihood of carrying forward compounds that are unlikely to perform well in further computational and experimental analyses. The 294 drug-like analogues will now undergo docking studies to evaluate their potential as inhibitors of *Sm*HMGR.

### 3.5. Comparison of docking algorithm performance using enrichment metrics

The performance of the three docking algorithms; AutoDock Vina, Glide SP, and Glide XP was evaluated based on their ability to distinguish between active inhibitors of HMGR and decoy compounds. The assessment was conducted using enrichment factor (EF) values at various top percentages of the ranked database, alongside the area under the receiver

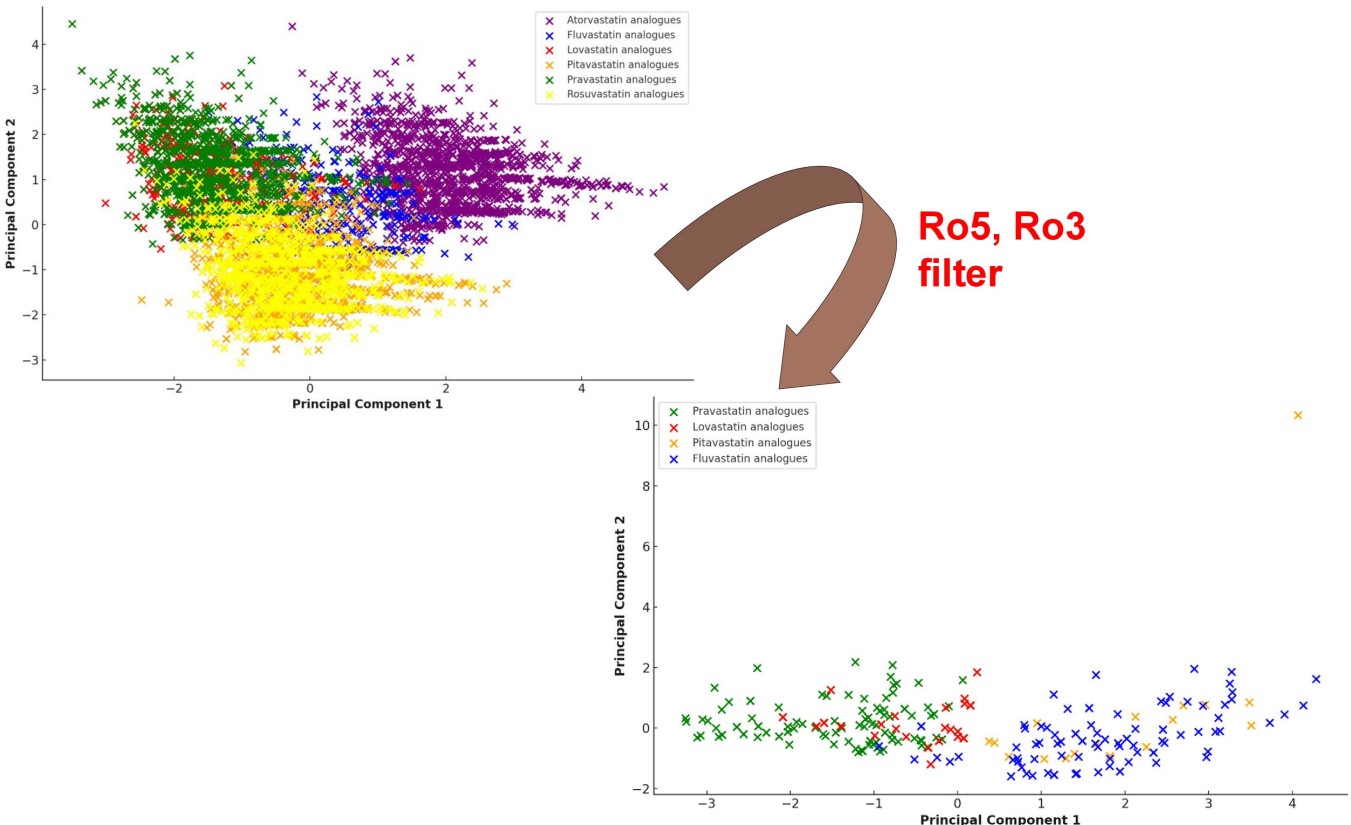

**Fig 5. Principal component analysis (PCA) of statin analogues based on physicochemical properties before and after Ro5 and Ro3 filtering: Fluvastatin analogues (blue), Pravastatin analogues (green), Lovastatin analogues (red), and Pitavastatin analogues (orange).** The PCA visualises the clustering of analogues according to molecular weight, LogP, number of rotatable bonds, hydrogen bond acceptors, and the number of rings.

**Table 3. Distribution of statin analogues: Drug-like, lead-like, and fragment-like properties.**

| Parent Statins | Total Analogues Found | Drug-like Analogues (Ro5) | Fragment-like Analogues (Ro3) | Lead-like Analogues |
|---|---|---|---|---|
| Atorvastatin | 2,118 | – | – | – |
| Rosuvastatin | 1,481 | – | – | – |
| Pravastatin | 1,157 | 143 | – | – |
| Pitavastatin | 3,718 | 23 | – | – |
| Fluvastatin | 404 | 94 | – | – |
| Lovastatin | 2,081 | 34 | – | – |
| Total | 11,959 | 294 | | |

operating characteristic curve (AUC) and logarithmic AUC (logAUC), to provide a detailed understanding of early enrichment and overall ranking quality.

AutoDock Vina demonstrated relatively strong early enrichment capabilities. At the top 1% of the ranked database, the algorithm achieved an EF of 4.87, indicating it retrieved nearly five times more actives than expected by random chance. This enrichment remained high at 5% (EF = 4.29) and 10% (EF = 3.58), confirming the algorithm's strength in prioritising active compounds near the top of the list. As expected, the EF value converged to 1.0 at 100%, reflecting the theoretical

baseline where all compounds are included and enrichment over random is no longer measurable. Overall, AutoDock Vina achieved an AUC of 72.66% (equivalent to 0.73) and a logAUC of 29.02, indicating moderate overall performance. This makes AutoDock Vina a suitable option for early-stage screening, where a focus on high enrichment at the top-ranked ligands is crucial, although its precision decreases as the ranking expands.

Glide SP, in contrast, showed a slower start in early enrichment compared to AutoDock Vina. The EF at 1% was 2.42, indicating weaker early enrichment. However, Glide SP's performance improved as more of the database was considered, with an EF of 3.35 at 5% and 2.62 at 10%, demonstrating a more balanced distribution of actives across the ranked ligands. This algorithm achieved an AUC of 73.52% (equivalent to 0.74) and a logAUC of 28.47 indicating that it maintains a reasonable level of accuracy across the entire ranked dataset, though its lower EF at 1% suggests that Glide SP is better suited for broader virtual screening efforts rather than prioritizing the highest-ranked actives. Its consistent performance across the ranked dataset implies that Glide SP could be useful in high-throughput screening contexts, where the goal is to cover a larger portion of the ranked database rather than achieve high precision in early enrichment.

Fig 6 illustrates the EF trends for each algorithm across ranked percentages of the dataset. Glide XP, on the other hand, significantly outperformed both AutoDock Vina and Glide SP in terms of early enrichment. The EF at 1% was 16.93, far surpassing the other two algorithms and demonstrating Glide XP's exceptional ability to enrich actives at the very top of the ranked database. However, like AutoDock Vina, its EF values decreased as more compounds were considered, with an EF of 5.27 at 5% and 3.10 at 10%. Despite this decline in enrichment at higher percentages of the ranked database, Glide XP maintained superior overall performance, as reflected by an AUC of 75.97% (equivalent to 0.76) and a logAUC of 34.33, highlighting its strong early enrichment capabilities. These values suggest that Glide XP not only excels at early-stage enrichment but also maintains higher precision across a broader range of ranked ligands compared to the other two algorithms. The steep drop in EF values after 5% underscores Glide XP's strength in refining a *small* number of highly ranked compounds, making it a powerful tool for applications where the top 1–5% of actives are of primary interest.

In comparing the three algorithms, Glide XP clearly demonstrated the best early enrichment capabilities. Its EF at 1% was more than three times higher than AutoDock Vina and nearly seven times higher than Glide SP, indicating that it was

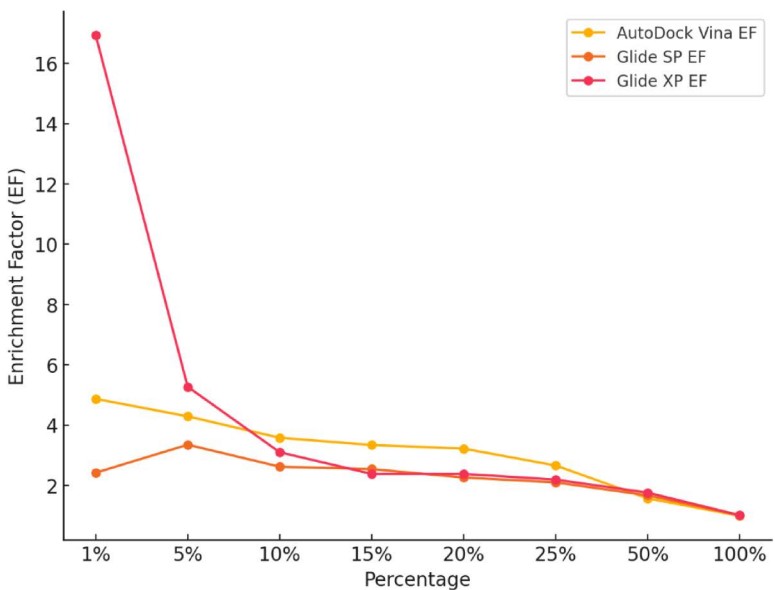

**Fig 6. Comparison of enrichment factors for AutoDock Vina, glide SP, and Glide XP across ranked percentages.**

the most effective at concentrating active ligands in the top-ranked positions. Furthermore, Glide XP's superior AUC and logAUC values reflect its overall precision and ability to accurately distinguish between actives and decoys across the ranked dataset (Fig 7). Ultimately, Glide XP's exceptional performance in early enrichment and its ability to accurately rank actives in the top 1% make it the best algorithm for docking statin analogues to *Sm*HMGR. Its precision in identifying highly active compounds ensures that it will be an effective tool for further exploration of statin analogues in this context, providing the highest likelihood of success in identifying potent inhibitors.

### 3.6. Binding affinity and structural analysis of statin and statin analogues docked to *Sm*HMGR active site

The results from the XP docking analysis provide detailed insights into the binding affinities of both the parent statins - fluvastatin, pravastatin, pitavastatin, and lovastatin—and their 294 statin analogues when docked into the active site of *Sm*HMGR (Fig 8). These scores, obtained through Glide XP docking, serve as a useful predictor for the potential inhibitory efficacy of compounds.

The docking analyses of four parent statins, namely pitavastatin, fluvastatin, pravastatin, and lovastatin, against the *Sm*HMGR active site revealed distinct binding affinities and led to the identification of structural analogues that met defined docking score thresholds. A total of 70 analogues were identified across the parent statins, all of which satisfied their respective cut-off criteria, as detailed alongside their structures in S1 Table.

Docking score thresholds were defined to identify analogues with binding affinities meeting or exceeding a predetermined level of predicted strength. Rather than applying a single uniform cut-off across all parent compounds, thresholds were tailored to each statin based on the docking score of the parent structure. This approach was adopted to account

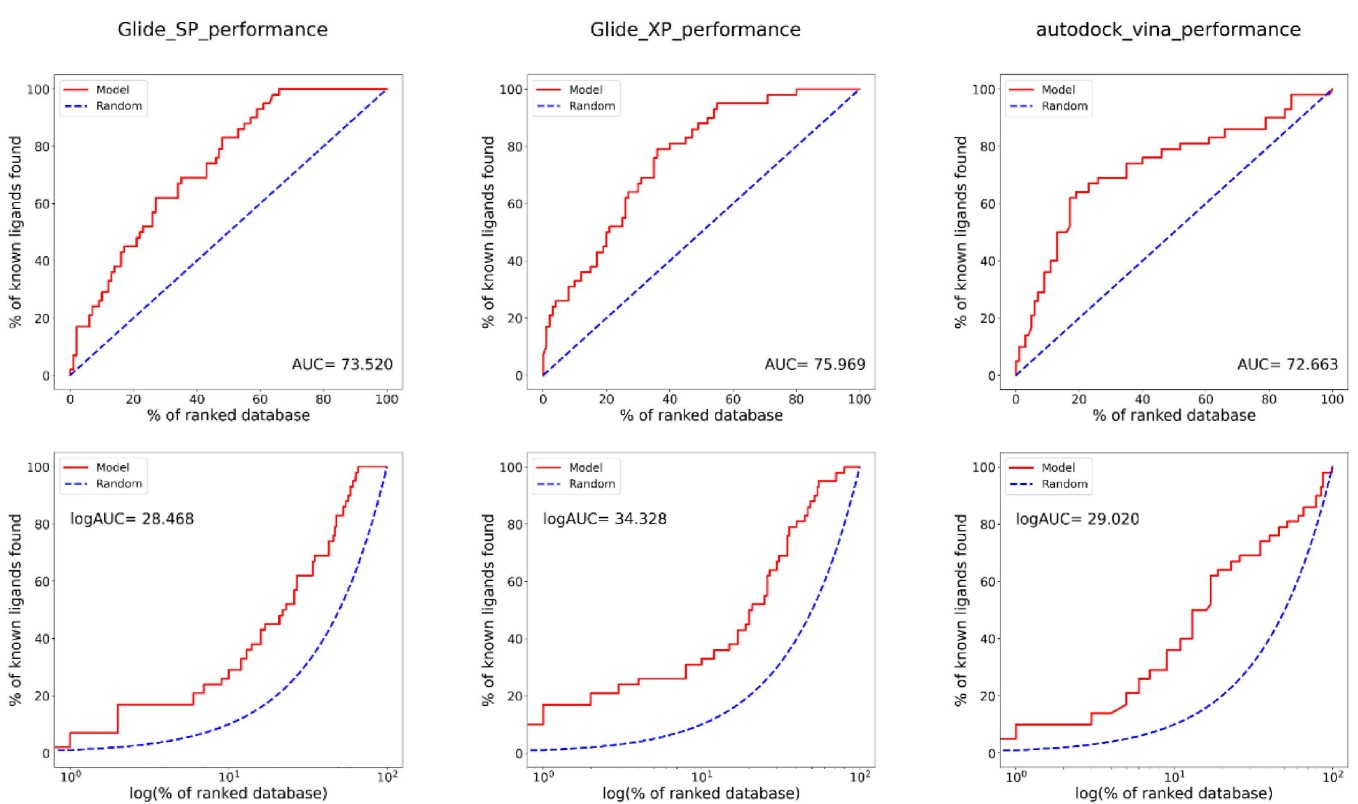

**Fig 7. Comparison of enrichment performance for AutoDock Vina, Glide SP, and Glide XP.**

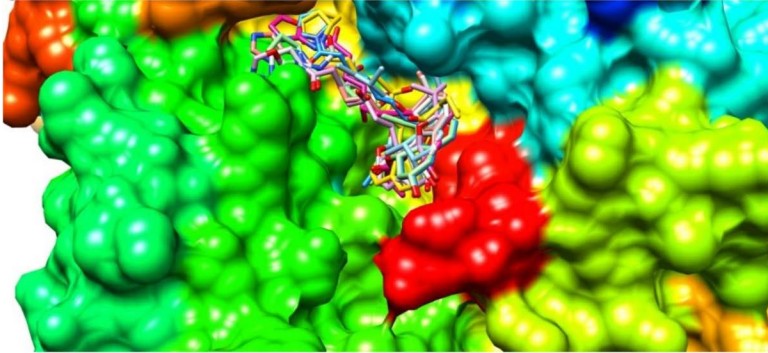

**Fig 8. Surface representation of SmHMGR with statin analogues docked in the active sites.**

for the inherent differences in the binding affinities of the parent statins, ensuring that analogues were evaluated relative to their structural baseline. Applying a universal threshold would risk unfairly excluding analogues derived from weaker-binding parents or over-representing those from stronger binders.

Pravastatin, which exhibited the strongest binding affinity (-10.886 kcal/mol), was assigned a cut-off of ≥ -7.5 kcal/mol, yielding 16 analogues (Table 4). Fluvastatin followed with a docking score of -9.442 kcal/mol, identifying 28 analogues above the same threshold. Pitavastatin, with a docking score of -8.420 kcal/mol, was evaluated using a slightly more lenient cut-off (≥ -7.0 kcal/mol), resulting in 14 analogues. Lovastatin, which showed the weakest binding (-7.11 kcal/mol), yielded 12 analogues that exceeded this same threshold. This parent-specific thresholding ensured a fair and meaningful selection process for analogue prioritisation.

To support the docking results, molecular dynamics (MD) simulations was carried out and the binding free energy (BFE) of *Sm*HMGR in complex with selected parent statins (lovastatin, pitavastatin, pravastatin, and fluvastatin) was estimated. The entire trajectory was considered to ensure accurate BFE calculations for both the parent compounds and their derivatives, and the BFE of all compounds is recorded in S1 Table. We selected analogues with comparable or greater binding free energy than their corresponding parent statins, focusing on the top four analogues for each statin class, except for fluvastatin and pitavastatin, which had three. The 2D structures of the top-performing analogues are presented below in Figs 9–12, along with their respective docking scores and binding free energies. The selected compounds were analyzed in-depth to explore their ligand-receptor non-covalent interactions (S1–S4 Figs), individual energy components contributing to their BFE, as well as their RMSD profiles (S5 Fig). This comprehensive analysis provided insights into the structural and energetic factors driving their binding efficiency. Subsequently, these compounds were advanced to experimental validation to confirm their binding performance and potential as effective *Sm*HMGR inhibitors.

**Table 4. Docking performance of statins and their analogues against XP docking score thresholds.**

| Statins | XP Docking Score (kcal/mol) | XP Docking Score Cut-offs (kcal/mol) | Number of Analogues Exceeding Cut-off |
|---|---|---|---|
| Fluvastatin | -9.442 | >= -7.5 | 28 |
| Pravastatin | -10.886 | >= -7.5 | 16 |
| Pitavastatin | -8.42 | >= -7.0 | 14 |
| Lovastatin | -7.11 | >= -7.0 | 12 |
| Total | | | 70 |

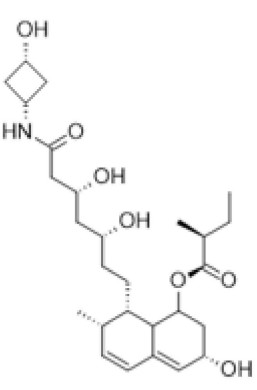

**Pravastatin**
-10.886 kcal/mol
-20.48 kcal/mol

**Analogue 1**
**Pravastatin Hydroxy-pyrrolidinamide**
-8.88 kcal/mol
-43.82 kcal/mol

**Analogue 2**
**Pravastatin 3-cis hydroxyoxetanylamide**
-8.24 kcal/mol
-31.65 kcal/mol

**Analogue 3**
**Pravastatin Pyrrolidinamide**
-7.89 kcal/mol
-33.40 kcal/mol

**Analogue 4**
**Pravastatin N-Methylcyclopropanamide**
-7.50 kcal/mol
-49.53 kcal/mol

**Fig 9. Docking scores (black) and binding free energies (red) of pravastatin and its top analogues at SmHMGR active sites.**

### 3.7. Comprehensive estimation of the components of the binding free energy

Binding free energy calculations provide a quantitative measure of the stability and affinity of ligand-protein complexes [62]. This analysis (Fig 13) underscores the energetic factors influencing the binding efficiency of *Sm*HMGR with statins and their analogues, revealing the contributions of van der Waals, electrostatic, and solvation effects in driving ligand affinity.

In the Lovastatin group, Lovastatin, the reference compound, exhibited a moderate binding free energy ($\Delta G_{bind}$ = -31.12 kcal/mol) with balanced contributions from van der Waals and electrostatic interactions (Table 5). Pravastatin Thiazolidinamide displayed a similar binding profile ($\Delta G_{bind}$ = -31.11 kcal/mol), indicating minimal impact from its modifications. In contrast, the addition of a 1,3,4-oxadiazol-2-yl-amide group improved binding affinity ($\Delta G_{bind}$ = -39.96 kcal/mol) by enhancing both dispersive and electrostatic interactions, despite a slight increase in solvation penalties. Among the analogues,

**Fig 10. Docking scores (black) and binding free energies (red) of pitavastatin and its top analogues at SmHMGR active sites.**

Pravastatin Isoxazol-5-yl-amide exhibited the strongest binding affinity ($\Delta G_{bind}$ = -50.9 kcal/mol), driven by powerful gas-phase interactions that compensated for its higher solvation cost (Table 5). Pravastatin-3 Pyrrolinamide also improved on the parent statin ($\Delta G_{bind}$ = -42.47 kcal/mol), primarily through strengthened van der Waals forces, though not to the same degree as the isoxazol derivative. These results highlight how structural variation can significantly enhance binding efficiency and interaction stability.

For Pitavastatin and its derivatives, structural variations also had a measurable impact on binding. The parent compound, Pitavastatin, showed a stable binding profile ($\Delta G_{bind}$ = -34.11 kcal/mol), supported by strong gas-phase interactions ($\Delta G_{gas}$ = -103.62 kcal/mol), with contributions from van der Waals ($\Delta E_{vdW}$ = -37.37 kcal/mol) and electrostatic forces ($\Delta Eele$ = -66.98 kcal/mol). Pitavastatin N-Methyl-2-methoxyethylamide showed slightly weaker binding ($\Delta G_{bind}$ = -31.09 kcal/mol), largely due to reduced electrostatic interactions despite slightly stronger dispersive forces (Table 6). In contrast, Pitavastatin N-Methyl-2-hydroxyethylamide demonstrated significantly stronger binding ($\Delta G_{bind}$ = -50.32 kcal/mol), driven by robust van der Waals ($\Delta E_{vdW}$ = -55.34 kcal/mol) and electrostatic ($\Delta E_{ele}$ = -35.95 kcal/mol) contributions, which outweighed moderate solvation costs. The N-Methyl-2-hydroxypropylamide derivative achieved intermediate binding ($\Delta G_{bind}$ = -34.74 kcal/mol) but higher solvation effects limited its potential relative to the hydroxyethylamide derivative.

**Fig 11. Docking scores (black) and binding free energies (red) of lovastatin and its top analogues at SmHMGR active sites.**

Pravastatin and its analogues revealed a diverse range of binding strengths. Pravastatin itself had modest binding affinity ($\Delta G_{bind}$ = -20.49 kcal/mol), dominated by electrostatic contributions ($\Delta E_{ele}$ = -134.71 kcal/mol) but weakened by significant solvation penalties (Table 7). Pravastatin Hydroxypyrrolidinamide showed a marked improvement ($\Delta G_{bind}$ = -43.82 kcal/mol) by achieving a favorable balance between dispersive forces ($\Delta E_{vdW}$ = -46.62 kcal/mol) and solvation. Pravastatin 3-cis Hydroxyoxetanylamide and Pravastatin Pyrrolidinamide had moderate binding affinities ($\Delta G_{bind}$ = -31.65 kcal/mol and $\Delta G_{bind}$ = -33.41 kcal/mol, respectively), benefiting from stronger dispersive forces but losing ground in electrostatics or solvation compatibility. Pravastatin N-Methylcyclopropanamide was the top performer ($\Delta G_{bind}$ = -49.43 kcal/mol), combining robust dispersive interactions with notable electrostatic forces and manageable solvation penalties.

Fluvastatin, the parent compound, displayed strong binding ($\Delta G_{bind}$ = -49.37 kcal/mol), supported by balanced van der Waals ($\Delta E_{vdW}$ = -42.18 kcal/mol) and electrostatic ($\Delta E_{ele}$ = -45.60 kcal/mol) contributions. Among its derivatives, Fluvastatin N-Hydroxyethylamide maintained competitive binding ($\Delta G_{bind}$ = -47.15 kcal/mol) due to stronger dispersive forces compensating for weaker electrostatics. In contrast, Fluvastatin N-Cyclohexylamine and N-Cyanoamide had reduced binding,

**Fig 12. Docking scores (black) and binding free energies (red) of fluvastatin and its top analogues at SmHMGR active sites.**

with less optimal interaction profiles and higher solvation effects (Table 8). This analysis confirms that while Fluvastatin remains a strong binder, its derivatives can vary widely in their performance depending on the balance of gas-phase and solvation energies. Altogether, these findings underscore the importance of structural modifications in tailoring binding interactions, demonstrating how specific changes can either enhance or weaken the performance of statins as effective *Sm*HMGR inhibitors.

### 3.8. Experimental validation of the identified statin analogues

To validate the computational findings, experimental testing was conducted using the top-performing statin analogues and their parent compounds. For reasons of synthetic accessibility, the analogue of Fluvastatin was excluded. Additionally, Pravastatin 3-cis hydroxyoxetanylamide and Pravastatin 1,3,4-oxadiazol-2-yl-amide were not included due to anticipated delays in synthesis. This resulted in a final selection of nine statin analogues for testing, alongside three parent compounds: Lovastatin, Pravastatin, and Pitavastatin as control. These compounds were screened for their activity against newly transformed schistosomula (NTS) and adult *S. mansoni* worms *in vitro*.

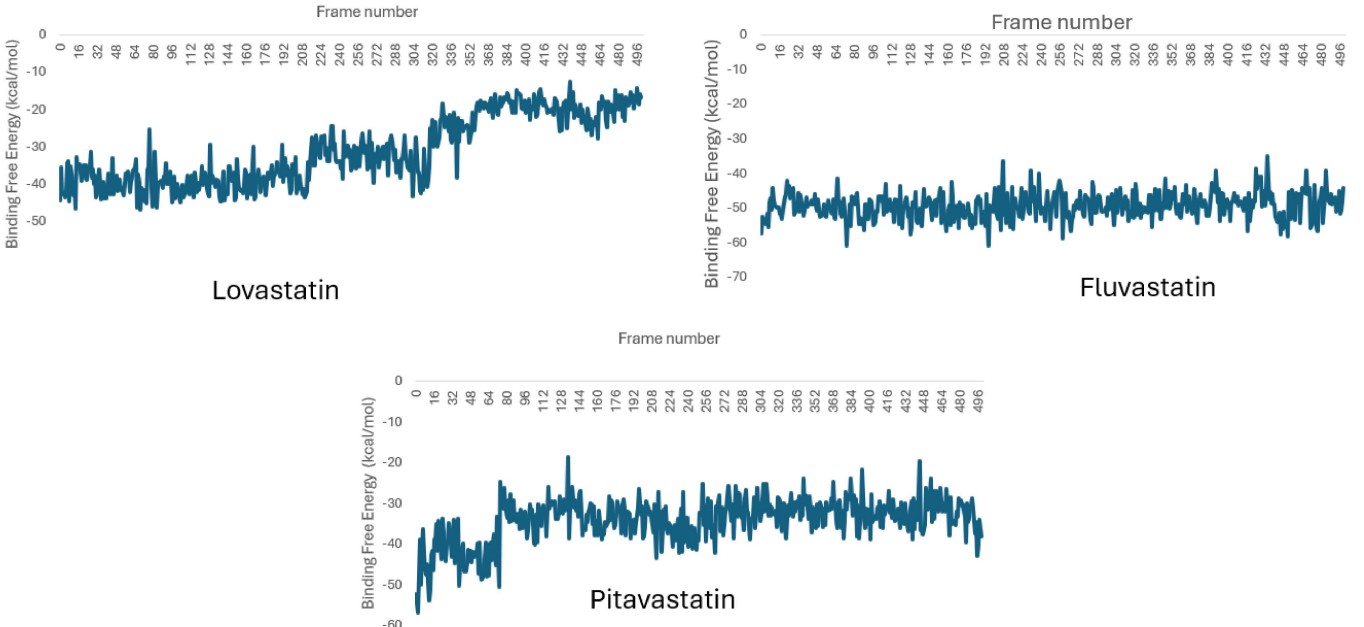

**Fig 13. Binding free energy profiles of Lovastatin, Fluvastatin, and Pitavastatin over 500 ns MD simulations.**

**Table 5. Energy component analysis of *Sm*HMGR complexes with Lovastatin and analogues.**

| Complexes | Energy Components (kcal/mol) | | | | | | |
|---|---|---|---|---|---|---|---|
| | $\Delta E_{vdW}$ | $\Delta E_{ele}$ | $\Delta G_{gas}$ | $\Delta G_{ele,sol(GB)}$ | $\Delta G_{np,sol}$ | $\Delta G_{sol}$ | $\Delta G_{bind}$ |
| *SmHMGR-Lovastatin* | -41.27 | -24.53 | -65.79 | 40.21 | -5.54 | 34.67 | -31.12 |
| *SmHMGR-Pravastatin Thiazolidinamide* | -44.51 | -22.59 | -67.1 | 42.22 | -6.22 | 35.99 | -31.11 |
| *SmHMGR-Pravastatin 1,3,4 -oxadiazol-2-yl-amide* | -49.25 | -47.22 | -96.47 | 56.51 | -6.89 | 56.51 | -39.96 |
| *SmHMGR-Pravastatin Isoxazol -5-yl-amide* | -58.61 | -49.65 | -108.27 | 65.91 | -8.55 | 57.36 | -50.9 |
| *SmHMGR-Pravastatin-3 Pyrrolinamide* | -51.82 | -26.5 | -78.32 | 42.97 | -7.12 | 35.85 | -42.47 |

### 3.8.1. Activity against newly transformed schistosomula *in vitro*.

The initial screening of statin and statin-like compounds at a concentration of 10 µM against *S. mansoni* NTS showed no significant activity, with none of the tested compounds achieving an effect ≥66% after 72 hours of incubation. Subsequently, the compounds were tested at a higher concentration of 50 µM, resulting in three hits (25% hit rate) that elicited effects exceeding ≥66% after 72 hours of incubation (Table 9). However, none of these hits were classified as lethal (effect = 100%), with the most active compound being Pitavastatin N-Methyl, -2-methoxyethylamide, which demonstrated 95.8% activity. The investigation of the schistosomicidal potential of three commercially available statins—lovastatin, pravastatin, and pitavastatin—revealed modest activity against schistosomula (NTS). At a concentration of 10 µM, lovastatin, pravastatin, and pitavastatin achieved NTS clearance rates of 31.3%, 37.5%, and 29.2%, respectively, after 72 hours of incubation. Increasing the concentration to 50 µM improved the clearance rates to 56.3%, 45.8%, and 39.6%, respectively. However, this increase in concentration did not result in a proportional enhancement of NTS clearance, suggesting that higher drug concentrations might not significantly improve efficacy against NTS.

**Table 6. Energy component analysis of *Sm*HMGR complexes with Pitavastatin and analogues.**

| Complexes | Energy Components (kcal/mol) | | | | | | |
|---|---|---|---|---|---|---|---|
| | $\Delta E_{vdW}$ | $\Delta E_{ele}$ | $\Delta G_{gas}$ | $\Delta G_{ele,sol(GB)}$ | $\Delta G_{np,sol}$ | $\Delta G_{sol}$ | $\Delta G_{bind}$ |
| *SmHMGR*-Pitavastatin | -37.37 | -66.98 | -103.62 | 75.47 | -5.96 | 69.51 | -34.11 |
| *SmHMGR*-Pitavastatin N-Methyl,-2-methoxyethylamide | -43.07 | -22.98 | -66.05 | 41.23 | -6.27 | 34.96 | -31.09 |
| *SmHMGR*-Pitavastatin N-Methyl-2-hydroxyethylamide | -55.34 | -35.95 | -91.29 | 48.92 | -7.95 | 40.97 | -50.32 |
| *SmHMGR*-Pitavastatin N-Methyl-2-hydroxypropylamide | -49.26 | -40.12 | -89.39 | 61.67 | -7.02 | 54.65 | -34.74 |

**Table 7. Energy component analysis of *Sm*HMGR complexes with Pravastatin and analogues.**

| Complexes | Energy Components (kcal/mol) | | | | | | |
|---|---|---|---|---|---|---|---|
| | $\Delta E_{vdW}$ | $\Delta E_{ele}$ | $\Delta G_{gas}$ | $\Delta G_{ele,sol(GB)}$ | $\Delta G_{np,sol}$ | $\Delta G_{sol}$ | $\Delta G_{bind}$ |
| *SmHMGR*-Pravastatin | -35.83 | -134.71 | -170.42 | 155.74 | -5.80 | 149.93 | -20.49 |
| *SmHMGR*-Pravastatin Hydroxy-pyrrolidinamide | -46.62 | -33.63 | -80.26 | 42.51 | -6.01 | 36.44 | -43.82 |
| *SmHMGR*-Pravastatin 3-cis hydroxyoxetanylamide | -42.67 | -39.32 | -81.99 | 56.55 | -6.21 | 50.33 | -31.65 |
| *SmHMGR*-Pravastatin Pyrrolidinamide | -48.72 | -18.73 | -67.45 | 40.73 | -6.68 | 34.05 | -33.41 |
| *SmHMGR*-Pravastatin N-Methylcyclopropanamide | -48.41 | -54.37 | -102.78 | 60.53 | -7.18 | 53.35 | -49.43 |

**Table 8. Energy component analysis of *Sm*HMGR complexes with Fluvastatin and analogues.**

| Complexes | Energy Components (kcal/mol) | | | | | | |
|---|---|---|---|---|---|---|---|
| | $\Delta E_{vdW}$ | $\Delta E_{ele}$ | $\Delta G_{gas}$ | $\Delta G_{ele,sol(GB)}$ | $\Delta G_{np,sol}$ | $\Delta G_{sol}$ | $\Delta G_{bind}$ |
| *SmHMGR*-Fluvastatin | -42.18 | -45.60 | -87.78 | 45.00 | -6.59 | 38.41 | -49.37 |
| *SmHMGR*-Fluvastatin N-Cyclohexylamine | -50.49 | -30.54 | -81.02 | 51.75 | -7.01 | 44.70 | -36.32 |
| *SmHMGR*-Fluvastatin N-Hydroxyethylamide | -56.53 | -25.13 | -81.66 | 42.43 | -7.93 | 34.51 | -47.15 |
| *SmHMGR*-Fluvastatin N-Cyanoamide | -38.53 | -50.27 | -88.79 | 57.26 | -6.19 | 51.06 | -37.74 |

**3.8.2. *In vitro* test activity on adult *S. mansoni*.** All three hits identified from the NTS activity screen (Pitavastatin (2R)-N-Methyl-2-hydroxypropylamide, Pitavastatin (1 R)-N-Methyl-2-hydroxyethylamide, and Pitavastatin N-Methyl, -2-methoxyethylamide) were further tested against adult *S. mansoni* at a concentration of 50 µM. All three compounds exhibited activity levels of 50–55% against adult worms under the same conditions (Table 9).

Interestingly, despite the increased concentration, no evidence of time-dependent clearance of NTS was observed for these statins. This finding contrasts with previous reports, particularly those by Rojo-Arreola *et al*. [19], which demonstrated time-dependent high efficacy of lovastatin. This discrepancy might be attributed to differences in experimental design, length of incubation of parasite with the drug, or variations in schistosome strains. The lack of compelling results for NTS led to the exclusion of lovastatin, pravastatin, and pitavastatin from further testing against adult schistosomes.

**Table 9. *In vitro* test analysis of statin and statin-analogues against S. *mansoni* (adults and newly transformed schistosomula).**

| CAT No | Product name | Dissolved | NTS<br>Effect in % (dead, 72h) and SD<br>Testconc. 50 µM | NTS<br>Effect in % (dead, 72h) and SD<br>Testconc. 10 µM | *S. mansoni* adult<br>Effect in % (dead, 72h) and SD<br>Testconc. 50 µM |
|---|---|---|---|---|---|
| P531070 | Pitavastatin (2R)-N-Methyl-2-hydroxypropylamide | DMSO | 70.8 (8.3) | 35.4 (6.2) | 50 (4.2) |
| P806530 | Pravastatin Pyrrolidine Amide | DMSO | 43.8 (2.1) | 35.4 (2.1) | |
| P806540 | Pravastatin N-Methylcyclopropanamine Amide | DMSO | 45.8 (4.2) | 33.3 (4.2) | |
| L472225 | Lovastatin | Methanol | 56.3 (6.3) | 31.3 (6.3) | |
| P531060 | Pitavastatin -N-Methyl-2-hydroxyethylamide | Methanol | 93.8 (2.1) | 27.1 (2.1) | 53.2 (1) |
| P531020 | Pitavastatin N-Methyl, -2-methoxyethylamide | Methanol | 95.8 (0) | 45.8 (0) | 55.3 (1) |
| P702000 | Pravastatin | DMSO | 45.8 (0) | 37.5 (0) | |
| P806510 | Pravastatin -Hydroxy-pyrrolidinamide (>90%) | DMSO | 43.8 (2.1) | 39.6 (2.1) | |
| P531000 | Pitavastatin | DMSO | 39.6 (2.1) | 29.2 (0) | |
| P806550 | Pravastatin Thiazolidinamide | DMSO | 43.8 (2.1) | 37.5 (4.2) | |
| P806580 | Pravastatin-3-Pyrrolinamide | DMSO | 47.9 (2.1) | 37.5 (4.2) | |
| P806520 | Pravastatin 3-cis-hydroxy-cyclobutylamide | DMSO | 25 (5.8) | 17.3 (1.9) | |

In contrast, the three analogues of pitavastatin (Table 9) exhibited promising activity against both NTS and adult schistosomes. This observation suggests that the pitavastatin analogues possess enhanced schistosomicidal properties compared to their parent compound. This result highlights the potential of structural modifications in improving the efficacy of statin-based compounds against schistosomes. These findings underscore the potential of certain statin analogues for further development as antischistosomal agents and emphasize the necessity of exploring their mechanisms of action to improve their therapeutic potential.

## 4. Conclusion

This study highlights the potential of computational and experimental methodologies to advance the discovery of potential inhibitors of *Schistosoma mansoni* HMGR (*Sm*HMGR). Homology modelling, molecular docking, and binding free energy analyses were used to identify and prioritise statin analogues for biological testing. While several statins, including pitavastatin, are commercially available, their schistosomicidal activities varied. The parent statins exhibited modest activity against schistosomula, whereas certain pitavastatin analogues demonstrated enhanced effects, including morphological alterations and impaired motility in adult worms. This study demonstrates how computational predictions can inform and complement experimental validation, accelerating the identification of promising drug candidates.

Although the *in silico* analysis was focused on identifying compounds with predicted high affinity for *Sm*HMGR, the *in vitro* assays employed here were phenotypic in nature and do not directly confirm target engagement. However, given that all tested compounds were close structural analogues of known statins originally developed for their activity against human HMGR, it is reasonable to expect a shared mechanism of action. As such, the improved schistosomicidal activity observed for several analogues is likely attributable to enhanced interaction with *Sm*HMGR rather than unrelated pharmacokinetic or off-target effects. This moderate alignment between computational predictions and phenotypic outcomes strengthens the rationale for further investigation of *Sm*HMGR as a drug target.

The findings contribute to a growing body of evidence supporting *Sm*HMGR as a viable therapeutic target for schistosomiasis. By integrating computational and experimental approaches, this research underscores the potential for rational drug design to overcome the limitations of existing treatments. Future studies should focus on further characterizing the mechanisms of action of these analogues, enhancing their pharmacological properties, evaluating their efficacy in

preclinical models and assessing selectivity to ensure minimal off-target effects on the human homologue. Ultimately, this work lays the foundation for novel therapeutic interventions that could significantly reduce the global burden of schistosomiasis.

While the computational pipeline identified analogues with favourable binding scores and predicted interactions against *Sm*HMGR, the modest activity observed in phenotypic assays highlights the known translational challenges of target-based screening. Factors such as permeability barriers, compound bioavailability, metabolic inactivation, and organismal complexity in S. *mansoni* may limit the manifestation of target-level predictions in whole-organism assays. The study's exploratory design sought to map the chemical space of statin analogues and assess their potential as *Sm*HMGR inhibitors, rather than optimise leads. We acknowledge the gap between *in silico* predictions and phenotypic outcomes and believe that bridging this gap will require further refinement of compound properties, including pharmacokinetics and selectivity in future studies.

## Supporting information

**S1 Table. Docking scores and binding free energy (MMGBSA) estimates of statins and their structural analogues grouped by parent compound.**
(DOCX)

**S1 Fig. 2D interaction diagrams of pitavastatin and its derivatives with *Sm*HMGR active site residues.**
(DOCX)

**S2 Fig. 2D Interaction Diagrams of Fluvastatin and Its Derivatives with *Sm*HMGR Active Site Residues.**
(DOCX)

**S3 Fig. 2D interaction diagrams of pravastatin and its derivatives with *Sm*HMGR active site residues.**
(DOCX)

**S4 Fig. 2D interaction diagrams of lovastatin and its derivatives with *Sm*HMGR active site residues.**
(DOCX)

**S5 Fig. Structural stability of *Sm*HMGR complexes with statins and their top derivatives.**
(DOCX)

## Acknowledgments

We appreciate the High-Performance Computing resources made available by the Centre for Biomolecular Sciences, School of Pharmacy, University of Nottingham, University Park, United Kingdom.

## Author contributions

**Conceptualization:** Kehinde F. Paul-Odeniran, Charles A. Laughton.

**Data curation:** Kehinde F. Paul-Odeniran, Paul O. Odeniran, Cécile Häberli.

**Formal analysis:** Kehinde F. Paul-Odeniran, Cécile Häberli, Charles A. Laughton.

**Funding acquisition:** Kehinde F. Paul-Odeniran.

**Investigation:** Kehinde F. Paul-Odeniran, Paul O. Odeniran, Cécile Häberli, Jennifer Keiser.

**Methodology:** Kehinde F. Paul-Odeniran, Paul O. Odeniran, Cécile Häberli, Jennifer Keiser, Charles A. Laughton.

**Project administration:** Kehinde F. Paul-Odeniran, Cécile Häberli, Jennifer Keiser, Charles A. Laughton.

**Resources:** Kehinde F. Paul-Odeniran, Cécile Häberli, Jennifer Keiser, Charles A. Laughton.

**Software:** Kehinde F. Paul-Odeniran, Charles A. Laughton.

**Supervision:** Jennifer Keiser, Charles A. Laughton.

**Validation:** Kehinde F. Paul-Odeniran, Paul O. Odeniran, Cécile Häberli, Jennifer Keiser, Charles A. Laughton.

**Visualization:** Kehinde F. Paul-Odeniran, Paul O. Odeniran, Jennifer Keiser, Charles A. Laughton.

**Writing – original draft:** Kehinde F. Paul-Odeniran.

**Writing – review & editing:** Kehinde F. Paul-Odeniran, Paul O. Odeniran, Cécile Häberli, Jennifer Keiser, Charles A. Laughton.

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
