## [Decision Letter · Decision Letter 0]

7 Jul 2025

Computational and Experimental Exploration of Statin and Statin-like Compounds as Potential Treatment of Schistosomiasis.

Dear Dr. PAUL-ODENIRAN,

Thank you for submitting your manuscript to PLOS Neglected Tropical Diseases. After careful consideration, we feel that it has merit but does not fully meet PLOS Neglected Tropical Diseases's publication criteria as it currently stands. Therefore, we invite you to submit a revised version of the manuscript that addresses the points raised during the review process.

Please submit your revised manuscript within 60 days Sep 05 2025 11:59PM. If you will need more time than this to complete your revisions, please reply to this message or contact the journal office at plosntds@plos.org. Please include the following items when submitting your revised manuscript:

We look forward to receiving your revised manuscript.

Kind regards,

David J. Diemert, M.D.

Academic Editor

Jong-Yil Chai

Section Editor

Shaden Kamhawi

co-Editor-in-Chief

Paul Brindley

co-Editor-in-Chief

**Journal Requirements:**

1) Please upload all main figures as separate Figure files in .tif or .eps format. For more information about how to convert and format your figure files please see our guidelines: 

2) We have noticed that you have uploaded Supporting Information files, but you have not included a list of legends. Please add a full list of legends for your Supporting Information files after the references list.

3) Please amend your detailed Financial Disclosure statement. This is published with the article. It must therefore be completed in full sentences and contain the exact wording you wish to be published.

2) If any authors received a salary from any of your funders, please state which authors and which funders..

4) We have amended your Competing Interest statement to comply with journal style. We kindly ask that you double check the statement and let us know if anything is incorrect. 

**Reviewers' Comments:**

Reviewer's Responses to Questions

**Key Review Criteria Required for Acceptance?**

**Methods**

-Are the objectives of the study clearly articulated with a clear testable hypothesis stated?

-Is the study design appropriate to address the stated objectives?

-Is the population clearly described and appropriate for the hypothesis being tested?

-Is the sample size sufficient to ensure adequate power to address the hypothesis being tested?

-Were correct statistical analysis used to support conclusions?

-Are there concerns about ethical or regulatory requirements being met?

Reviewer #1: The objective of the study was to use a computational pipeline to systematically identify and optimize SmHMGR inhibitors. The computational part of the study, although extensive, lacks objectivity and it is reflected on the modest antischistosomal activities found for the final compounds selected by the computational pipeline. The computational methodology is based on structure-based drug design tools but the experimental validation is performed by phenotypic assays. This causes an important gap between the computational predictions and computational results. Authors didn't discuss this important aspect of their study. Are such modest experimental findings the result of poor performance of the computational methodology or the inherent hurdles in translating target-based predictions to in vitro biological activities on whole organisms?

Other specific points for consideration:

- Authors should define “activity” early on the manuscript. It seems form pieces of information scattered across the manuscript that “activity” is the percentage of dead parasites. In this case, authors should clearly define how parasite death was assessed.

- Topic 2.1: What changes from each of the 140 models to the other?

- Topic 2.2: Is atorvastatin a known ligand of SmHMGR. What makes it a good template to search for other inhibitors of the enzyme?

- Line 128: “atorvastatin and each SmHMGR model were optimally formatted in .pdb files”. What exactly this means? What were the preparation steps?

- Lines 197-198: Please clarify. What do the authors mean by "stereochemical variability"? How 10 stereoisomers can be generated per ligand?

- Lines 203-204: Specify what algorithm was used with what restraints, number of minimization steps and convergence criteria.

Reviewer #2: Minor questions and comments to be addressed by authors in the attached file.

**Results**

-Does the analysis presented match the analysis plan?

-Are the results clearly and completely presented?

-Are the figures (Tables, Images) of sufficient quality for clarity?

Reviewer #1: Topic 3.2 (Inverse docking): Are there any published methodological papers supporting this claim? I'm not convinced of the utility of this approach. The major flaw for me is that you are using a docking protocol, which has not been validated to select target models.

Minor issues:

- Line 353 and 390: Define "strong". Please use RMSD metric. Define "robust" and "stable", authors should try to be as objective as possible using quantitative strucutral parameters to support their observations instead of vague adjectives.

- Fig 1: The last G (human) and D (SCHMA) are highlighted but are distinct. There is also a V-L pair between 240 and 300

-Line 403: What is rec061?

Reviewer #2: Minor questions and comments to be addressed by authors in the attached file.

**Conclusions**

-Are the conclusions supported by the data presented?

-Are the limitations of analysis clearly described?

-Do the authors discuss how these data can be helpful to advance our understanding of the topic under study?

-Is public health relevance addressed?

Reviewer #1: The conclusions are, for the most part, not supported by the data. For instance, authors state in line 774 that "analogues of pitavastatin (Table 9) exhibited promising activity". Based on what premises? What is the target activity/potency profile for an initial hit in schistosomiasis drug discovery?

Authors also claim to have "optimized" statin analogues but optimization is the process where one start from an initial hit and then optimize the chemical structure to improve biological activities. But activity of the parent compound was not known.

Reviewer #2: Minor questions and comments to be addressed by authors in the attached file.

**Editorial and Data Presentation Modifications?**

Reviewer #1: Overall, authors use a large part of the manuscript to describe computational results that ended up not performing as well as expected in terms of experimental validation. The manuscript would gain a lot in readability if the authors made an effort to only described the most relevant results and the ones directly relating to the experimental results. Everything else should be move to supplementary material.

Reviewer #2: Minor questions and comments to be addressed by authors in the attached file.

**Summary and General Comments**

Reviewer #1: In Summary, this study aimed to employ structure-based drug design computational tools to identify new inhibitors of SmHMGR, a rising target for antischistosomal drug discovery. The study is greatly limited by the fact that experimental validation of the results are obtained straight from phenotipc assays on in vitro cultivated parasites without first testing on the molecular target. Althogh this approach is feasible, authors seem to completely ignore it in their discussion and make overstatements about the performance of the computational approach. In my view there is interesting data woth publishing in this manuscript but should be majorly reorganized to become more concise and focused on the discussion of the experimental results in view of the limitations of the computational approach.

Minor issues:

Introduction:

Line 65: authors should more accurately discuss this PZQ limitations since the latter is actually able to kill schistosomula in vitro at least and consider the juvenile stage as well.

Lines 80-81: Discuss similarity with human homologue and discuss potential selectivity issues.

Line 85: How many statins have been found to actually inhibit SmHMGR? A figure identifying the most potent ones would be relevant in the introduction.

Reviewer #2: Minor questions and comments to be addressed by authors in the attached file.

PLOS authors have the option to publish the peer review history of their article (what does this mean? ). If published, this will include your full peer review and any attached files.

**Do you want your identity to be public for this peer review?** For information about this choice, including consent withdrawal, please see our Privacy Policy .

Reviewer #1: No

Reviewer #2: No

**Figure resubmission:**

**Reproducibility:**



---

## [Decision Letter · Decision Letter 1]

2 Sep 2025

Dear Dr PAUL-ODENIRAN,

We are pleased to inform you that your manuscript 'Computational and Experimental Exploration of Statin and Statin-like Compounds as Potential Treatment of Schistosomiasis.' has been provisionally accepted for publication in PLOS Neglected Tropical Diseases.

Best regards,

David J. Diemert, M.D.

Academic Editor

Jong-Yil Chai

Section Editor

Shaden Kamhawi

co-Editor-in-Chief

Paul Brindley

co-Editor-in-Chief

Reviewer #1:

Reviewer #2:

Reviewer's Responses to Questions

**Key Review Criteria Required for Acceptance?**

**Methods**

-Are the objectives of the study clearly articulated with a clear testable hypothesis stated?

-Is the study design appropriate to address the stated objectives?

-Is the population clearly described and appropriate for the hypothesis being tested?

-Is the sample size sufficient to ensure adequate power to address the hypothesis being tested?

-Were correct statistical analysis used to support conclusions?

-Are there concerns about ethical or regulatory requirements being met?

Reviewer #1: (No Response)

Reviewer #2: (No Response)

**Results**

-Does the analysis presented match the analysis plan?

-Are the results clearly and completely presented?

-Are the figures (Tables, Images) of sufficient quality for clarity?

Reviewer #1: (No Response)

Reviewer #2: (No Response)

**Conclusions**

-Are the conclusions supported by the data presented?

-Are the limitations of analysis clearly described?

-Do the authors discuss how these data can be helpful to advance our understanding of the topic under study?

-Is public health relevance addressed?

Reviewer #1: (No Response)

Reviewer #2: (No Response)

**Editorial and Data Presentation Modifications?**

Reviewer #1: (No Response)

Reviewer #2: (No Response)

**Summary and General Comments**

Reviewer #1: I'm satisfied with the changes and comments provided by the authors.

Reviewer #2: (No Response)

PLOS authors have the option to publish the peer review history of their article (what does this mean? ). If published, this will include your full peer review and any attached files.

**Do you want your identity to be public for this peer review?** For information about this choice, including consent withdrawal, please see our Privacy Policy .

Reviewer #1: No

Reviewer #2: No

---

## [Editor Report · Acceptance letter]

Dear Dr PAUL-ODENIRAN,

We are delighted to inform you that your manuscript, " 

Computational and Experimental Exploration of Statin and Statin-like Compounds as Potential Treatment of Schistosomiasis.," has been formally accepted for publication in PLOS Neglected Tropical Diseases.

Best regards,

Shaden Kamhawi

co-Editor-in-Chief

Paul Brindley

co-Editor-in-Chief
